# Olfactory Impairment Correlates with Executive Functions Disorders and Other Specific Cognitive Dysfunctions in Parkinson’s Disease

**DOI:** 10.3390/biology12010112

**Published:** 2023-01-10

**Authors:** Paolo Solla, Carla Masala, Tommaso Ercoli, Claudia Frau, Caterina Bagella, Ilenia Pinna, Francesco Loy, Giovanni Defazio

**Affiliations:** 1Neurological Unit, AOU Sassari, University of Sassari, Viale S. Pietro 10, 07100 Sassari, Italy; 2Department of Biomedical Sciences, University of Cagliari, SP 8 Cittadella Universitaria, 09042 Monserrato, Italy; 3Institute of Neurology, Azienda Ospedaliero Universitaria di Cagliari, SS 554 km 4.500, 09042 Cagliari, Italy

**Keywords:** Parkinson’s disease, olfactory dysfunction, cognitive impairment, executive function

## Abstract

**Simple Summary:**

Olfactory and cognitive disorders represent important non-motor symptoms in Parkinson’s disease. No clear evidence has been reported about the association of each specific cognitive domain and olfactory impairment in patients with Parkinson’s disease. This study aims to evaluate associations between olfactory dysfunction and specific cognitive domains in patients compared to controls. Our data suggested a significant association between olfactory dysfunction and deficit in executive functions.

**Abstract:**

Introduction. Olfactory and cognitive disorders represent important non-motor symptoms in Parkinson’s disease (PD). No clear evidence was reported about association of specific cognitive domains and olfactory impairment. Objective. The aim of this study was to evaluate the association between olfactory dysfunction and specific cognitive domains in PD patients compared to controls. Methods. 178 PD patients and 98 controls were enrolled and evaluated for odor threshold (OT), discrimination (OD), identification (OI), and TDI score using the Sniffin’ Sticks test. Cognitive function was evaluated using the Montreal Cognitive Assessment scale with six sub-scores: Orientation (OIS), Attention (AIS), Language (LIS), Visuospatial (VIS), Memory (MIS), and Executive index scores (EIS). Results. Statistically significant correlations were observed between OT versus, LIS, and between TDI score versus EIS. Multivariate linear regression analysis, including age and sex which are well-known predictors of olfactory dysfunction, showed that, among specific cognitive domains, only LIS was significant predictor for OT, VIS was a significant predictor for OD, while both EIS and AIS were significant predictors for OI, and finally only EIS was significant predictor for TDI score. Conclusions. Olfactory disorders in PD patients appear commonly related to dysfunction of specific cognitive domains, with strict association between global olfactory impairment and executive function deficits.

## 1. Introduction

Olfactory impairment represents one of the earliest and most common non-motor symptoms in Parkinson’s disease (PD) [1,2] with up to 95% of PD patients affected by olfactory deficits [3,4]. Cognitive impairment is also a common non-motor disorder in PD patients and may progress in dementia during the course of the disease in a substantial number of patients with a point prevalence of dementia close to 30% [5]. Moreover, non-motor symptoms such as hyposmia, depression, and sleep behavior disorders are closely associated with cognitive decline, and the presence of these symptoms may predict the subsequent development of PD dementia [6,7].

Although previous studies evaluated the correlation between olfactory deficit and cognitive impairment in PD [8,9,10,11,12,13] demonstrating that olfactory dysfunction may increase the risk of dementia up to ten years after PD diagnosis regardless of baseline cognitive function [14], the involved specific cognitive domains are largely undetermined. To date, only a single study explored whether Mild Cognitive Impairment (MCI) and the involvement of each single cognitive domain may influence olfactory function in PD patients [15].

The aim of this study was to evaluate the association between odor threshold (OT), odor discrimination (OD), and odor identification (OI), with specific cognitive domains in PD patients compared to controls.

## 2. Materials and Methods

### 2.1. Participants

In this study, 276 participants were enrolled (149 men and 127 women), including 178 PD patients (mean age ± SD; 70.2 ± 9.3) and 98 controls (mean age ± SD; 68.4 ± 8.0). Consecutive PD outpatients were recruited during regular out-patient follow-up visits at the Movement Disorders Center of the University of Cagliari and gave their written informed consent to participate in the study. PD was diagnosed according to the Movement Disorder Society Clinical Diagnostic Criteria for PD [16] and performed by a neurologist specialized in movement disorders. Controls were healthy subjects attending neurology outpatient clinics for a routine check-up and had no history of PD or any other neurodegenerative disease.

In all participants exclusion criteria were any disorder interfering with olfactory evaluation, such as chronic/acute rhinosinusitis, stroke, history of head or neck trauma, atypical parkinsonism, and psychiatric conditions.

### 2.2. Procedures

Demographic and clinical information for each participant included sex, age, weight, height, and body mass index (BMI). In PD patients, additional data regarding current medications and age at PD onset were collected. All PD patients were in a stable ON condition and assessments were carried out in all recruited patients after receiving their usual medication. Motor impairment was assessed by the Unified PD Rating Scale (UPDRS) part III [17] and disability with the Modified Hoehn and Yahr (HY) Stage [18]. The levodopa equivalent daily dose (LEDD) was computed as previously reported [19]. PD patients and healthy controls underwent an olfactory evaluation with the Sniffin’ Sticks Extended test (SSET) [20]. SSET is an assessment of olfactory chemosensory function which consists of tests for OT, OD, and OI, with established reliability and validity [21]. Sniffin’ Sticks are pen-like odor-dispensing devices. Each pen with a length of 14 cm and an inner diameter of 1.3 cm was positioned at around 2 cm in front of both participants’ nostrils for few seconds. Participants could drink only water 1 h before the experiment and were instructed to avoid smoking and scented products during the testing day. All subjects were blindfolded for the OT and OD tasks [22]. According to SSET Guideline, was initially assessed OT using n-butanol with 16 stepwise dilutions [23]. The OT was evaluated using the single-staircase technique and a three-alternative forced-choice task (3AFC). The OT scores may range from 16 for participants who were able to detect the lowest concentration of the n-butanol to 1 when participants who were unable to detect its highest concentration. Second, OD evaluation was performed using 16 trials. In the OD test, three different pens were presented using the 3AFC task, two containing the same odor and the third containing the target odorant. The OD score was calculated as the sum of correct responses and may range from 0 to 16 points. Third, OI evaluation was assessed with the use of 16 common odors presented with four verbal descriptors in a multiple forced choice format (one target and three distractors). The total score (threshold–discrimination–identification: TDI) was calculated. TDI values as >30.5, ≤30.5 and ≤16.5 were considered as normosmia, hyposmia, and functional anosmia, respectively.

For each participant cognitive abilities were assessed with the Montreal Cognitive Assessment (MoCA), which consists of different domains: visual-constructional skills, executive functions, attention and concentration, memory, language, conceptual thinking, calculations, and spatial orientation [24,25]. The MoCA scale is a test commonly used in cognitive screening for detection of MCI with a sensitivity of 90% and a specificity of 87% for detecting subjects with MCI and distinguishing them from subjects with normal cognition [25]. According to Julayanont and colleagues [26], we calculated 6 index sub-scores of MoCA representative of specific domains of cognitive dysfunction: Orientation Index Score (OIS), Attention Index Score (AIS), Language Index Score (LIS), Visuospatial Index Score (VIS), Memory Index Score (MIS), and Executive Index Score (EIS).

### 2.3. Data Analysis

Analysis was conducted using SPSS 26.0 for Windows (IBM, Armonk, NY, USA). All data were presented as mean values ± standard deviation (SD). Statistical differences between PD patients and control groups for all variables were assessed by means of independent sample *t* test adjusted with Bonferroni correction for multiple comparisons, or the Yates-corrected chi-square test, as appropriate. In order to identify the more promising factors for the multivariate regression analyses, correlations among each specific domain of cognitive function and olfactory impairment were performed using Pearson’s correlation coefficient (*r*) with Bonferroni correction for multiple comparisons. Moreover, an exploratory stepwise multivariate linear regression analysis was performed to assess the potential contribution of each significant correlated factor (OIS, AIS, LIS, VIS, MIS, and EIS) on the olfactory function in PD patients. The multivariate linear regression analysis was performed using OT, OD, OI, and TDI score as dependent variables in different models, while the 6 index sub-scores of MoCA representative of specific domains of cognitive dysfunction (OIS, AIS, LIS, VIS, MIS, and EIS) were independent variables. In order to perform the multivariate linear regression analysis using a stepwise selection, in the model 1, we calculated the correlation between OT versus EIS and LIS as independent variables; then in model 2, we included the OD as dependent variable versus VIS as independent variable; in model 3, we used OI as a dependent variable versus EIS; finally, in the model 4 TDI score we used TDI score as a dependent variable versus EIS. Each model was also adjusted by age and sex. This stepwise method allows us to evaluate the specific role of each independent variable in the model. *p* values < 0.05 were considered statistically significant.

### 2.4. Ethical Considerations

The study was conducted according to the 1964 Declaration of Helsinki and its later amendments, and with the guidelines of The Ethics Committees of the AOU Cagliari. All involved subjects gave their written informed consent and received an explanatory statement to participate in the study.

## 3. Results

For all participants demographic and clinical features are reported in the Table 1.

Subjects with PD and controls were similar for weight, height, age, and BMI. In PD patients, the mean of the disease duration was 4.2 ± 3.6 years, H&Y mean score was 2.1 ± 0.8, and mean of the UPDRS pars-III score was 20.8 ± 12.5. All assessed parameters related to the olfactory function were statistically significant different between PD patients and controls. Indeed, comparing PD patients with controls, mean scores of OT, OD, and OI scores were 2.6 ± 2.2 versus 5.7 ± 4.3 (*p* ≤ 0.001), 7.5 ± 3.1 versus 10.5 ± 2.7 (*p* ≤ 0.001), and 7.6 ± 3.3 vs. 11.7 ± 2.5 (*p* ≤ 0.001), respectively. Furthermore, PD patients showed a significant decrease in TDI mean scores compared to controls (17.7 ± 6.9 vs. 28.0 ± 7.1, *p* ≤ 0.001, respectively).

Evaluating global cognitive assessment, PD patients showed a significant impairment in MoCA mean scores compared to controls (21.1 ± 5.1 versus 25.7 ± 3.2, *p* ≤ 0.001).

Sub-scores of different MoCA indexes of PD patients and controls, representative of specific domains of cognitive dysfunction, are reported in Table 2. All MoCA indexes were statistically significant different between PD patients and controls, with AIS, LIS, VIS, MIS, and EIS showing *p* ≤ 0.001, while in OIS *p* was ≤0.008 (Table 2).

In Table 3, significant bivariate correlations between parameters of olfactory function assessed in PD patients and specific index sub-scores of MoCA are reported. We found statistically significant correlations between OT versus LIS score (*r* = 0.245, *p* = 0.001) (Table 3). The other correlations for OD and OI versus each specific index sub-scores of MoCA did not reach the level of significance after Bonferroni correction for multiple comparisons. Moreover, we found statistically significant correlation between TDI versus EIS score (*r* = 0.206, *p* < 0.006).

Furthermore, to better clarify the impact of these bivariate correlations, multivariate linear regression analyses were performed to predict olfactory dysfunction in PD patients, after correction for age and sex, in relation to specific index sub-scores of MoCA, using OT, OD, OI, and TDI as dependent variables (Table 4).

Multivariate linear regression analysis showed that only LIS and sex were significant predictors using OT as a dependent variable [F_(1,176)_ = 11.274, *p* < 0.001]. The models explained around the 5% of the variance for the LIS (R^2^ = 0.055) (Figure 1A) and around the 7% of the variance for the age and sex (R^2^ = 0.071). Using the OD score as a dependent variable only VIS, age, and sex were significant predictors [F_(1,176)_ = 6.090, *p* < 0.015]. The models explained around the 3% of the variance (R^2^ = 0.028) for the VIS (Figure 1B) and around the 7% of the variance for the age and sex (R^2^ = 0.074); while for OI score both EIS, AIS, age, and sex were significant predictors [F_(1,176)_ = 6.933, *p* < 0.001]. Models explained around the 5% of the variance for EIS and AIS (R^2^ = 0.046) (Figure 1C,D) and around the 12% of the variance for the age and sex (R^2^ = 0.118).

Finally, multivariate linear regression analysis showed that using TDI score as a dependent variable only EIS, age and sex were significant predictors [F_(1,176)_ = 7.782, *p* < 0.006]. The model explained around the 4% of the variance for the EIS (R^2^ = 0.037) (Figure 1E) and 12% of the variance for age and sex (R^2^ = 0.123).

## 4. Discussion

Both olfactory deficits and cognitive impairment represent well known non-motor disorders in PD [1,5]; moreover, the cognitive impairment may progress in dementia during the course of the disease in a substantial number of patients [5,27]. In this context, a recent study of Fang and colleagues [11] showed more severe anosmia in PD patients with cognitive impairment using University of Pennsylvania Smell Identification Test (UPSIT) and the MoCA scale. Our data indicated that PD patients showed a significant impairment for each specific domain of cognitive function in the index sub-scores of MoCA compared to controls. However, olfactory deficits are age-related and are usually associated to mild cognitive impairment in older adults [28,29,30]. In particular, these previous studies [28,29,30] suggested that olfactory dysfunction in older age is associated with impaired global cognition and a neurocognitive profile characterized by more rapid decline in memory, attention, and perceptual processing speed. Previous studies demonstrated that olfactory functions were related to the specific brain areas (e.g., entorhinal cortex, occipital cortex, intraparietal sulcus, and insula) involved in chemosensory processing [31,32]. In this study we evaluated the correlation between olfactory dysfunction and specific cognitive domains in PD patients. Interestingly, we found significant bivariate correlations between the olfactory threshold versus the language score domain and between TDI score, which is the sum of the main components involved in olfaction, and the specific index sub-score of MoCA related to executive function. Indeed, the OT is usually associated to the nasal epithelium and individual differences of the nasal cavity [23], while the odor discrimination and the odor identification are usually associated to central pathways such as the orbitofrontal cortex, the piriform cortex, and the amygdala [33].

Although previous studies evaluated and demonstrated the association between olfactory disorders and cognitive impairment in PD [8,9,10,11,12,13,14,15], only few studies explored if the olfactory dysfunction in PD may be related to a specific cognitive domain [15].

Thus, these findings seem to suggest that olfactory dysfunction in PD may be dissimilar among the different main components of olfactory impairment with peculiar patterns involving specific cognitive domains. However, the impairment of executive functions seems to be involved in the global burden of olfactory disturbances and might represent a peculiar cognitive “biomarker”. These considerations are in agreement with previous studies demonstrating that in humans, the olfactory function plays an important role on prefrontal-dependent cognitive functions, almost certainly by common cerebral circuits [33].

Indeed, executive functions, which are collectively referred to executive function and cognitive control, are higher order cognitive capacities that allow people to orient towards the future, display self-control and effectively have goal-oriented behavior [34]. Thus, the correlation between global olfactory function (TDI score) and deficit in executive functions in PD patients of our study suggests a common neural pathway involved in olfactory processing and executive functions. Many of these functions are located in the frontal lobes [35,36,37,38]. In particular, the study of Chaudhary and colleagues [37] suggested that an altered energy metabolism, an impaired cholinergic neuronal transmission, and the neuronal function loss may be implicated in the frontal lobe pathology of PD patients.

Among these considerations, our results also support the hypothesis that the olfactory threshold, more associated with sensorial processing, is not clearly associated to executive functions while can be related with specific cognitive domain such as the language function. Indeed, while OT is considered a predominant sensorial process mainly depending on the peripheral and subcortical part of the olfactory system, OI and OD are, respectively, the expression of the ability to identify and differentiate between odorants [34].

To confirm these data and better clarify the impact of these correlations, we performed a multivariate linear regression analysis aimed to identify which of the specific cognitive domains was the most significant predictor of the olfactory dysfunction in PD patients. This multivariate analysis, including also age and sex which are well-known independent predictors of olfactory disturbances [28,29,39,40,41,42,43], confirmed that, among specific cognitive domains, only the deficit of executive functions was a significant predictor of the global olfactory impairment, using TDI as a dependent variable.

Finally, this analysis showed that only LIS and VIS dysfunctions were respectively significant predictors for threshold and discrimination, while both EIS and AIS functions were significant predictors for the impairment in odor identification. The association of visuospatial dysfunction with the olfactory impairment is not surprising, also bearing in mind a previous study which identified that olfaction was worse in PD patients with visuospatial dysfunction [15]. In particular, previous studies suggested that PD patients showed impaired verbal learning, memory, and executive function associated to lower olfactory function [44,45].

Although our research was carried out in a large population of PD patients and controls, some limitations should be cited. First, this work has been designed as a cross-sectional study and, thus, did not report longitudinal data. Second, the MoCA has been used as a cognitive screening while a detailed cognitive assessment was not performed. Thus, this study might not capture all the complex aspects of cognitive functioning. However, previous studies have indicated that all MoCA index scores provide highly valid information on the function of each cognitive domain in patients with mild cognitive impairment and dementia, with the exception for the memory domain, which does not reflect the severity of memory impairment in patients affected by dementia [46,47]. Despite these limitations, the study provided clear data on the association between olfactory dysfunction and specific cognitive domains in PD patients.

## 5. Conclusions

Our data highlights a strict association between the global olfactory impairment and deficit in executive functions in PD patients. Under this scenario, it can be hypothesized that neural pathways involved in olfactory processing may extensively overlap with pathways contributing to executive functioning. Future studies should integrate the assessment of PD patients with a complete neuropsychological evaluation and with functional brain neuroimaging in order to better understand the relationship between specific cognitive domains and olfactory impairment.

## Figures and Tables

**Figure 1 biology-12-00112-f001:**
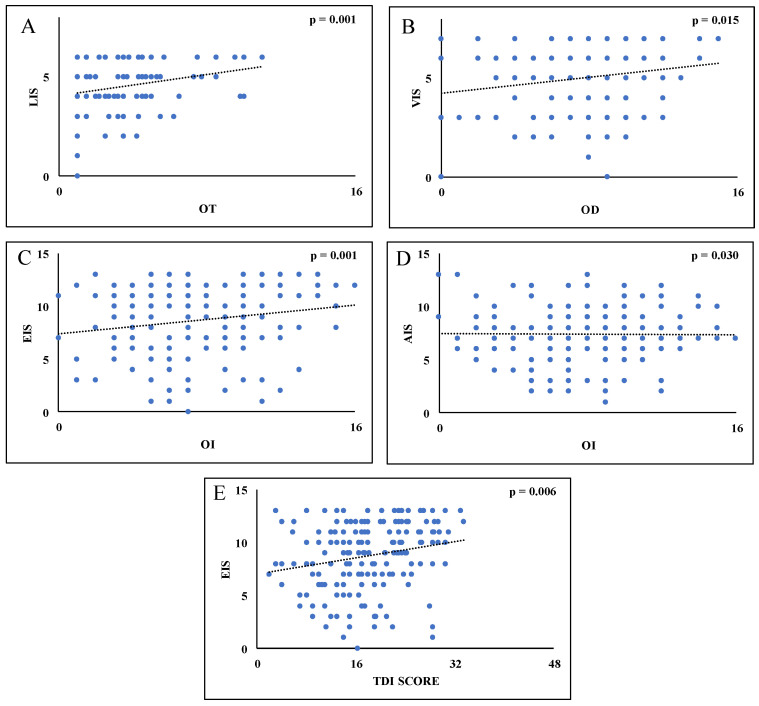
Scatterplots of the relationship between OT versus LIS (**A**), between OD versus VIS (**B**), between OI versus EIS (**C**), between OI versus AIS (**D**), and finally between TDI score versus EIS (**E**).

**Table 1 biology-12-00112-t001:** Clinical and demographic information of all participants. Data are expressed as mean ± SD.

Demographics	PD	Control	*p* Value
	N = 178	N = 98	
Sex N (% female)	75 (%42.1)	52 (%53.1)	0.081
Age	70.2 ± 9.3	68.4 ± 8.0	0.099
Weight (kg)	71.7 ± 15.9	70.1 ± 15.0	0.412
Height (cm)	164 ± 0.01	164 ± 0.1	0.999
BMI	26.6 ± 4.9	25.9 ± 4.1	0.344
PD duration (years)	4.2 ± 3.6	NA	NA
UPDRS	2.1 ± 0.8	NA	NA
LEDD	353 ± 293	NA	NA
HY	20.8 ± 12.5	NA	NA
OT	2.6 ± 2.2	5.7 ± 4.3	**0.001**
OD	7.5 ± 3.1	10.5 ± 2.7	**0.001**
OI	7.6 ± 3.3	11.7 ± 2.5	**0.001**
TDI	17.7 ± 6.9	28.0 ± 7.1	**0.001**
MoCA	21.1 ± 5.2	25.7 ± 3.2	**0.001**

Legend: PD, Parkinson’s disease; SD = standard deviation; BMI = body mass index; UPDRS-III = Unified PD Rating Scale part III; LEDD = levodopa equivalent daily dose; HY = Hoehn and Yahr stage; OT = odor threshold; OD = odor discrimination, OI = odor identification; TDI = threshold–discrimination–identification; MoCA = Montreal Cognitive Assessment. Significant *p* values are highlighted in bold. NA = not available.

**Table 2 biology-12-00112-t002:** Index sub-scores of MoCA representative of specific domains of cognitive dysfunction in PD patients and controls. Data are expressed as mean ± SD.

	Groups	N	Mean ± SD	*p* Value
OIS	PD	178	5.7 ± 0.8	**0.008**
	Controls	98	6.0 ± 0.2	
AIS	PD	178	7.4 ± 2.4	**0.001**
	Controls	98	9.3 ± 2.1	
LIS	PD	178	4.3 ± 1.2	**0.001**
	Controls	98	5.1 ± 0.9	
VIS	PD	178	4.9 ± 1.7	**0.001**
	Controls	98	6.4 ± 1.0	
MIS	PD	178	1.6 ± 1.6	**0.001**
	Controls	98	2.3 ± 1.6	
EIS	PD	178	8.7 ± 3.2	**0.001**
	Controls	98	11.5 ± 1.7	

Legend: AIS = Attention Index Score; EIS = Executive Index Score; LIS = Language Index Score; MIS = Memory Index Score; MoCA = Montreal Cognitive Assessment; OIS = Orientation Index Score; PD = Parkinson’s disease; VIS = Visuospatial Index Score. Significant *p* values are highlighted in bold.

**Table 3 biology-12-00112-t003:** Correlations between parameters of olfactory dysfunction (OT, OD, OI and TDI) and specific index sub-scores of MoCA in patients affected by Parkinson’s disease. Significance was set at the 0.0083 level after Bonferroni correction.

		OIS	AIS	LIS	VIS	MIS	EIS
OT	*r*	−0.009	0.193	0.245	0.181	0.099	0.161
	*p*	0.097	0.010	**0.001**	0.016	0.188	0.031
OD	*r*	0.103	0.002	0.153	0.183	0.112	0.152
	*p*	0.171	0.978	0.041	0.015	0.136	0.043
OI	*r*	0.130	−0.009	0.086	0.114	0.113	0.176
	*p*	0.084	0.906	0.254	0.128	0.134	0.019
TDI	*r*	0.106	0.061	0.192	0.196	0.136	0.206
	*p*	0.158	0.417	0.01	0.009	0.069	**0.006**

Legend: AIS = Attention Index Score; EIS = Executive Index Score; LIS = Language Index Score; MIS = Memory Index Score; MoCA = Montreal Cognitive Assessment; OIS = Orientation Index Score; PD = Parkinson’s disease; VIS = Visuospatial Index Score. Significant *p* values are highlighted in bold.

**Table 4 biology-12-00112-t004:** Multiple linear regression analysis models using odor discrimination (OD), odor identification (OI), odor threshold (OT), TDI score as dependent variables and each significant correlated factor of MoCA sub-scores, age, and sex as independent variables.

	Unstandardized Coefficients		Standardized Coefficients		
	B	Std Error	β	*t*	*p*
Model 1: OT as a dependent variable					
LIS	0.454	0.135	0.245	3.358	**0.001**
Age	−0.021	0.019	−0.084	−1.087	**0.279**
Sex	0.700	0.338	0.152	2.070	**0.040**
Model 2: OD as a dependent variable					
VIS	0.331	0.134	0.183	2.468	**0.015**
Age	−0.059	0.026	−0.174	−2.259	**0.025**
Sex	1.222	0.463	0.193	2.637	**0.009**
Model 3: OI as a dependent variable					
EIS	0.323	0.100	0.311	3.238	**0.001**
AIS	−0.286	0.131	−0.210	−2.182	**0.030**
Age	−0.083	0.027	−0.231	−3.090	**0.002**
Sex	1.421	0.483	0.211	2.943	**0.004**
Model 4: TDI Score as a dependent variable					
EIS	0.444	0.159	0.206	2.790	**0.006**
Age	−0.176	0.056	−0.236	−3.166	**0.002**
Sex	3.365	0.999	0.240	3.368	**0.001**

Legend: AIS = Attention Index Score; EIS = Executive Index Score; LIS = Language Index Score; MIS = Memory Index Score; MoCA = Montreal Cognitive Assessment; OIS = Orientation Index Score; PD = Parkinson’s disease; VIS = Visuospatial Index Score. Significant *p* values are highlighted in bold.

## Data Availability

Not applicable.

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
