# Peer review of "Olfactory Impairment Correlates with Executive Functions Disorders and Other Specific Cognitive Dysfunctions in Parkinson’s Disease"

_biology, 2023, doi:10.3390/biology12010112_

Round 1

Reviewer 1 Report

The purpose of the study by Solla et al. was to investigate the possible association between the different components of the olfactory function and specific cognitive domains in Parkinson’s disease (PD) patients compared to healthy controls.  Subjects were evaluated for odor threshold, discrimination, identification, and the global olfactory score (TDI) by using the Sniffin’ Sticks Extended test. The cognitive abilities were studied with the Montreal Cognitive Assessment (MoCA) with six subscores: Orientation, attention, language, visuospatial, memory, and executive index.

Using a multivariate linear regression analysis, the authors show that: a) language score was predictor for odor threshold, b) visuospatial score was predictor for olfactory discrimination, c) executive index and attention score were predictor for olfactory identification, and d) executive index was predictor for TDI score. They conclude that in PD patients, there is a strict association between olfactory impairment and executive function deficits.

Although the study is of interest, several points need to be addressed:

1. In the abstract there are several mistakes regarding the correlation between olfactory components and cognitive scores when compared with the results described in Table 3.

2. Since the executive index is just correlated with the identification component of the olfactory function, the discussion and the conclusion of the manuscript need to be focused to this finding and no to overall olfactory impairment.

3. The discussion is poor, there are many repeated concepts from the introduction, methodology and results sections. However, the specific findings, the possible mechanisms involved and the comparative with previous publications are not well developed. There are a lot of published reports regarding these issues that are not discussed. For example, Schlintl and Schienle 2022. Exp Aging Res; Yap et al. 2022 Front Aging Neurosci; Tan et al., 2022 Front Aging Neurosci, between others.

Author Response

The purpose of the study by Solla et al. was to investigate the possible association between the different components of the olfactory function and specific cognitive domains in Parkinson’s disease (PD) patients compared to healthy controls.  Subjects were evaluated for odor threshold, discrimination, identification, and the global olfactory score (TDI) by using the Sniffin’ Sticks Extended test. The cognitive abilities were studied with the Montreal Cognitive Assessment (MoCA) with six subscores: Orientation, attention, language, visuospatial, memory, and executive index.

Using a multivariate linear regression analysis, the authors show that: a) language score was predictor for odor threshold, b) visuospatial score was predictor for olfactory discrimination, c) executive index and attention score were predictor for olfactory identification, and d) executive index was predictor for TDI score. They conclude that in PD patients, there is a strict association between olfactory impairment and executive function deficits.

Although the study is of interest, several points need to be addressed:

  1. In the abstract there are several mistakes regarding the correlation between olfactory components and cognitive scores when compared with the results described in Table 3.

Answer 1) Authors thank the Reviewer and have revised the Abstract. All corrections are in red color on the text.

  1. Since the executive index is just correlated with the identification component of the olfactory function, the discussion and the conclusion of the manuscript need to be focused to this finding and no to overall olfactory impairment.

Answer 2) Authors thank the Reviewer and have revised the Discussion and the Conclusion in agreement with these suggestions.

  1. The discussion is poor, there are many repeated concepts from the introduction, methodology and results sections. However, the specific findings, the possible mechanisms involved and the comparative with previous publications are not well developed. There are a lot of published reports regarding these issues that are not discussed. For example, Schlintl and Schienle 2022. Exp Aging Res; Yap et al. 2022 Front Aging Neurosci; Tan et al., 2022 Front Aging Neurosci, between others.

Answer 3) Authors thank the Reviewer for this suggestion and have revised all Discussion section and have added all these suggested references on the manuscript.

Reviewer 2 Report

The manuscript by Solla and co-authors studied correlaitons between olfactory functions and specific cognitive (dys)functions in patients with Parkinson's disease. Their results demonstrate correlation of olfactory dysfunction with decreased ability in specific cognitive domains. The manuscript is well-written, the method are clearly described and the results are well discussed. Overall the work is valuable to forward the determination of PD patients by olfactory loss and cognitive dysfunction in specific domains, and distinguish PD patients with other neurodegenerative disease with Olfactory dysfunction. I have the following comments for improvement.

1. Although the gender and age are not statistically different betweent PD and control groups, the could be included as covariables in the analyses models.

2. For correlation analyses (Table3), I would suggest the author do a correction for multiple comparisons.

Author Response

The manuscript by Solla and co-authors studied correlations between olfactory functions and specific cognitive (dys)functions in patients with Parkinson's disease. Their results demonstrate correlation of olfactory dysfunction with decreased ability in specific cognitive domains. The manuscript is well-written, the method is clearly described and the results are well discussed. Overall the work is valuable to forward the determination of PD patients by olfactory loss and cognitive dysfunction in specific domains, and distinguish PD patients with other neurodegenerative disease with Olfactory dysfunction. I have the following comments for improvement.

  1. Although the gender and age are not statistically different between PD and control groups, the could be included as covariables in the analyses models.

Answer 1) Authors thank the Reviewer and revised the multivariate linear regression analyses using age and sex as covariables in the models. Consequently, the Table 4 and Results section are now revised. All corrections are in red color on the text.

  1. For correlation analyses (Table3), I would suggest the author do a correction for multiple comparisons.

Answer 2) Authors thank the Reviewer and revised the correlation analyses using the Bonferroni correction for multiple comparisons. The Table 3, Abstract, and Results section are now revised.